# Background selection and biased gene conversion affect more than 95% of the human genome and bias demographic inferences

Fanny Pouyet[1,2†*], Simon Aeschbacher[1,2,3†], Alexandre Thiéry[1,2], Laurent Excoffier[1,2*]

[1]Computational and Molecular Population Genetics, Institute of Ecology and Evolution, University of Bern, Bern, Switzerland; [2]Swiss Institute of Bioinformatics, Lausanne, Switzerland; [3]Department of Evolutionary Biology and Environmental Studies, University of Zurich, Zurich, Switzerland

**Abstract** Disentangling the effect on genomic diversity of natural selection from that of demography is notoriously difficult, but necessary to properly reconstruct the history of species. Here, we use high-quality human genomic data to show that purifying selection at linked sites (i.e. background selection, BGS) and GC-biased gene conversion (gBGC) together affect as much as 95% of the variants of our genome. We find that the magnitude and relative importance of BGS and gBGC are largely determined by variation in recombination rate and base composition. Importantly, synonymous sites and non-transcribed regions are also affected, albeit to different degrees. Their use for demographic inference can lead to strong biases. However, by conditioning on genomic regions with recombination rates above 1.5 cM/Mb and mutation types (C⟷G, A⟷T), we identify a set of SNPs that is mostly unaffected by BGS or gBGC, and that avoids these biases in the reconstruction of human history.
DOI: https://doi.org/10.7554/eLife.36317.001

*For correspondence:
fanny.pouyet@gmail.com (FP);
laurent.excoffier@iee.unibe.ch (LE)

†These authors contributed equally to this work

Competing interests: The authors declare that no competing interests exist.

## Introduction

Human genomic diversity has evolved under diverse and complex constraints (*Auton et al., 2015*), such as past demography, selection, mutations, or genomic rearrangements (*Lohmueller et al., 2011*; *Schiffels and Durbin, 2014*; *Sudmant et al., 2015*; *Mallick et al., 2016*). However, the influence of these evolutionary forces and their interactions remain to be fully understood. For instance, it is yet unclear which fraction of the genome evolves under positive or purifying selection (*McVicker et al., 2009*; *Rands et al., 2014*; *Corbett-Detig et al., 2015*). Such information is crucial to our understanding of what portion of the genome is evolving neutrally, and necessary to form a clear basis for demographic inference, the detection of selective events, or the inference of the distribution of fitness effects of new mutations.

Genome-wide variation in recombination may strongly affect neutral variants (*Spencer et al., 2006*; *Corbett-Detig et al., 2015*), as selection will have more impact on linked polymorphism in regions of low recombination (*Charlesworth et al., 1995*), whereas biased gene conversion, which can also mimic the effect of selection (*Galtier and Duret, 2007*; *Ratnakumar et al., 2010*), will occur mostly in regions of high recombination (*Katzman et al., 2011*). In humans, various measures of diversity are positively correlated with levels of recombination (*Nachman, 2001*; *Spencer et al., 2006*; *Cai et al., 2009*; *Lohmueller et al., 2011*). While a direct mutagenic effect of recombination seems unlikely (*McVicker et al., 2009*; *Schaibley et al., 2013*) except at CpG sites

**eLife digest** Human chromosomes are made up of DNA, which contains about 3 billion 'letters' that carry the instructions needed to build and maintain an individual. However, only about 10 percent of the human genome is made up of genes that code for proteins, or have a defined role in the body. The DNA sequence is largely the same in all people, but some modifications – or variants – occur about every hundred letters. These produce different versions of the same gene, which give us our unique features, such as the color of our hair or eyes.

The frequencies of some genetic variants can change over time, which makes human populations diverge genetically and physically. This can happen through different mechanisms. Positive selection keeps variants that are beneficial in specific environments, while negative selection removes genetic changes that are detrimental, for example because they cause disease. Transmission bias favors one of the two variants from our two parents. Chance alters the frequencies of neutral variants, which are neither good nor bad for the individual.

It is important to distinguish between these different scenarios, as they inform us about the forces that act on human evolution. For example, neutral variants tell us about the demography and migration patterns between populations. Variants under negative selection reveal which genetic areas are under pressure to stay the same because they are important for the organism to function correctly. Until now, it was unclear how we could best identify the variants affected by different evolutionary pressures, and how much of the genome was under negative selection.

Pouyet, Aeschbacher et al. created a measure of genetic diversity that is only affected by selection or transmission bias. The results showed that negative selection influences as much as 85 percent of our genome, whereas transmission bias affects a majority of the rest of the genome. After removing these two biases, less than 5 percent of the human genome is found to evolve by chance. This suggests that while most of our genetic material is formed of non-functional sequences, the vast majority of it evolves indirectly under some type of selection.

These findings define which parts of our genome evolves neutrally and can therefore be used to correctly reconstruct the past demography and migration events of humans around the world. The next step could be to reassess the history of human populations that was drawn using genomic data.
DOI: https://doi.org/10.7554/eLife.36317.002

(*Arbeithuber et al., 2015*), there is still some debate about whether the correlation between diversity and recombination is driven by recurrent selective sweeps (hitchhiking of neutral and slightly deleterious mutations) or background selection (BGS; i.e. purifying selection against deleterious mutations at linked sites) (*McVicker et al., 2009*; *Stephan, 2010*; *Hernandez et al., 2011*; *Lohmueller et al., 2011*). The modeling of genomic diversity under selection in humans suggests that it can be explained entirely by BGS (*Lohmueller et al., 2011*), whereas a combination of both BGS and positive selection seems to best explain genomic diversity in *Drosophila* (*Elyashiv et al., 2016*). However, the correlation between diversity and recombination is generally relatively weak in humans for most tested statistics and seems restricted to genomic regions of relatively low-recombination rate (<1 cM/Mb, (*Cai et al., 2009*; *Lohmueller et al., 2011*)).

Given the positive relationship between recombination and genetic variability, it has been proposed that the genomic regions most suitable for demographic inferences should be far away from genes and have high-recombination rates (*Lohmueller et al., 2011*). However, regions of high recombination might be prone to GC-biased gene conversion (gBGC), a process by which GC alleles in recombination tracts are preferentially transmitted in GC/AT heterozygotes (*Duret and Galtier, 2009*). This process thus increases the frequency of G and C derived alleles (usually denoted as strong or S alleles, *Lachance and Tishkoff, 2014*) relative to A and T (denoted as weak or W alleles), especially in recombination hotspots (*Spencer et al., 2006*; *Glémin et al., 2015*). By modifying allele frequencies in high-recombination regions, gBGC affects the site frequency spectrum (SFS) (*Lachance and Tishkoff, 2014*; *Glémin et al., 2015*) such that it becomes right-shifted for W-to-S (WS) mutations and left-shifted for S-to-W (SW) mutations. In addition, gBGC affects various classical statistics used to detect selection, and WS SNPs show larger levels of population differentiation than other SNPs (*Lachance and Tishkoff, 2014*). Overall, gBGC is believed to directly affect only 1% to

2% of the human genome, near recombination hotspots (*Glémin et al., 2015*), but due to the transient nature of these hotspots, a larger fraction of the genome could have been affected in the long term.

Here, we use two whole-genome human datasets to determine how and to what extent recombination and selective forces affect genome-wide diversity in humans. We examine the relationship between recombination rate and the average derived allele frequency per individual, as well as the SFS. After determining the parts of the genome that are least affected by BGS and gBGC, we examine the impact of these two processes on the SFS, and how they affect demographic inference based on the SFS.

## Results

We first used a representative set of one hundred individuals from the 1000 Genomes (1000G) Project (*Auton et al., 2015*) from ten populations in five geographic regions to study the pattern of human genomic diversity. Since our analyses compared genomic diversity across individuals for sets of sites devoid of any missing data, we selected in each population those 1000G individuals with the highest coverage. As a measure of genomic diversity, we used the average derived allele frequency per individual ($\overline{DAF_i}$). This statistic was computed over all sites that were found polymorphic across all populations (i.e. where derived alleles are neither fixed nor absent in all individuals). Assuming that there are $S_{Tot}$ such sites, $\overline{DAF_i}$ is computed for each diploid individual as the total number of derived alleles observed at those sites ($n_i$) divided by 2 $S_{Tot}$. We show in the Materials and methods below how this statistic depends on the average time to the most recent common ancestor ($t_{MRCA}$) of the whole sample, and, if one assumes neutrality that this statistic should be the same on expectation for any individual in the sample across its whole genome, irrespective of the particular demography of its population (*Figure 1—figure supplement 1*). Differences in the number of derived alleles ($n_i$) among individuals and among genomic regions should therefore only reflect differences in selection, mutation rate and/or generation time (*Figure 1—figure supplement 2*). The number of derived alleles is indeed broadly comparable across individuals from different geographic regions, even though Southern and Eastern Asians (SAS and EAS) show a slight yet significant deficit in the number of derived alleles than the three other groups (~50,000 out of 17 million, Tukey test, p<0.01) (*Figure 1—figure supplement 2*), suggesting either a more efficient selection, a lower mutation rate or a longer generation time. This statistic thus appears ideally suited to assess the impact of selection at linked sites that could locally alter their $t_{MRCA}$.

### The average derived allele frequency per individual increases with recombination rate

For individuals belonging to five geographic regions, we studied the variability of $\overline{DAF_i}$ across the genome by computing it separately for SNPs that belong to different recombination classes and averaging it across individuals within each region (*Figure 1A*). Local recombination rates around each SNP were obtained from the 1000G Yoruba recombination map (*Frazer et al., 2007*) (see Materials and methods), but the use of alternative recombination maps leads to similar patterns (*Figure 1—figure supplement 3*). We find that the average intra-bin $\overline{DAF_i}$ increases almost log-linearly with the average recombination rate. The exception is for the lowest recombination class, most likely because low-recombination rates are difficult to estimate (*Kong et al., 2010*; *Wegmann et al., 2011*). We observe the same log-linear relationship in a set of 20 individuals chosen to represent five continents that were sequenced as part of the Simons Genome Diversity Project (*Mallick et al., 2016*) at higher coverage (31–60×) than the 1000G individuals (*Figure 1—figure supplement 4A*). The log-linear relationship between recombination rate and $\overline{DAF_i}$ is conserved among geographic regions (*Figure 1A*) and it is also observed at the level of single individuals (*Figure 1—figure supplement 5A*), as expected from our theoretical derivations. Note that this very similar behavior among individuals and populations is not in line with a differential action of positive selection (selective sweeps) in different continents. Therefore, if adaptive events were involved in shaping allele frequencies and creating this relation, they should have occurred before the human lineage split into different continental groups. Since most variation in exonic regions has emerged in the last 10,000 years (*Fu et al., 2013*), a pure adaptive explanation for this relation seems unlikely.

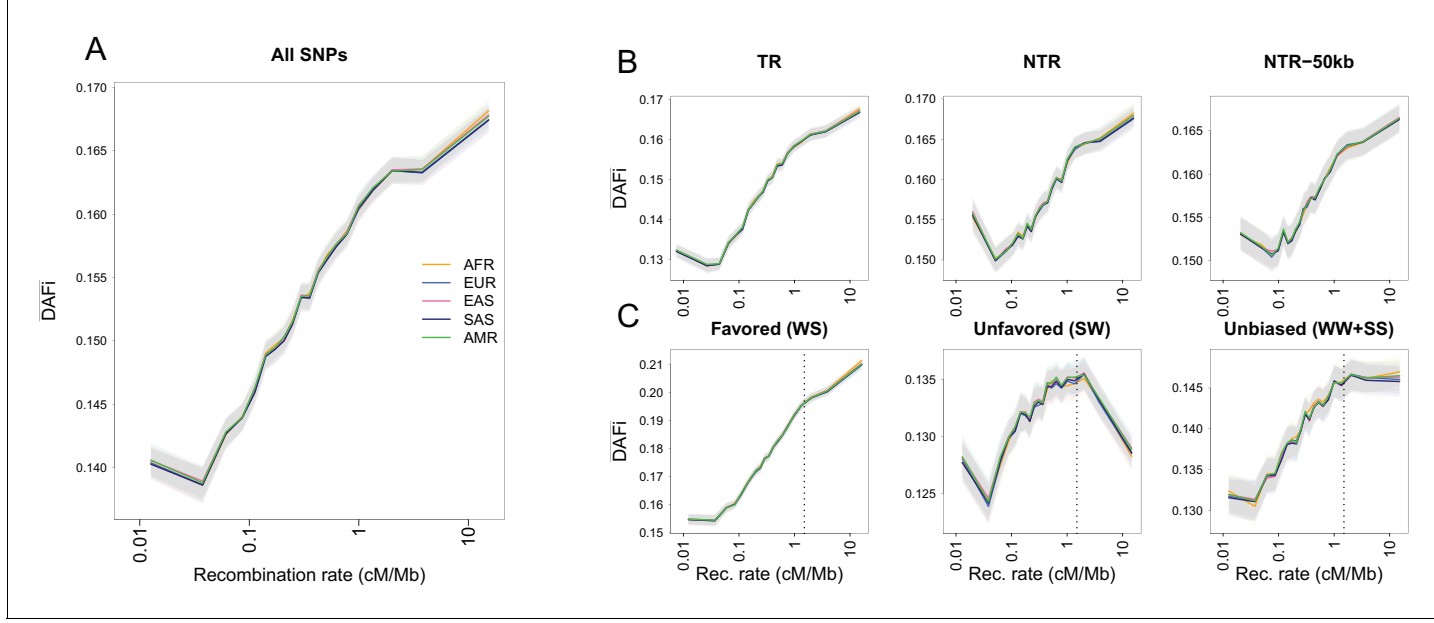

**Figure 1.** Average derived allele frequency per individual ($\overline{DAF_i}$) as a function of recombination rate. 1000G SNPs were ranked by their local recombination rate and divided into 20 bins of equal size. $\overline{DAF_i}$ was computed for each individual as the number of heterozygous sites plus two times the number of derived homozygous sites and averaged per geographic region. (**A**) $\overline{DAF_i}$ vs. recombination rate on a $\log_{10}$ scale for all 17,129,351 1000G SNPs. (**B**) Same as panel A for SNPs in transcribed regions (TR), non-transcribed regions (NTR), or non-transcribed regions more than 50 kb away from TR (NTR-50kb). (**C**) Same as panel A for SNPs differently affected by GC-biased gene conversion (gBGC). Left: WS sites, where the derived allele is favored by gBGC. Center: SW sites, where the ancestral allele is favored by gBGC. Right: WW and SS sites, which are not affected by gBGC. The vertical dashed lines at 1.5 cM/Mb delimit an approximate threshold above which BGS has no effect on WW and SS sites, but where gBGC has a strong and opposite effect on WS and SW sites. Each group (AFR: Africans, EUR: Europeans, EAS: East-Asians, SAS: South Asians, AMR: Admixed Americans) includes individuals from two populations (see **Supplementary file 1** - Table S1). Shaded areas delimit the 95% confidence interval of each group, estimated using a block-bootstrap approach (see Materials and methods).

DOI: https://doi.org/10.7554/eLife.36317.003

The following figure supplements are available for figure 1:

**Figure supplement 1.** Genealogy of a sample of two diploid individuals at an arbitrary locus.

DOI: https://doi.org/10.7554/eLife.36317.004

**Figure supplement 2.** Individual number of derived alleles ($n_i$) for 1000G individuals.

DOI: https://doi.org/10.7554/eLife.36317.005

**Figure supplement 3.** The increase of $\overline{DAF_i}$ with recombination rate is robust to the choice of the recombination map.

DOI: https://doi.org/10.7554/eLife.36317.006

**Figure supplement 4.** Same Figure and legend as **Figure 1** but based on 20 SGDP individuals sequenced at high-coverage (**Mallick et al., 2016**).

DOI: https://doi.org/10.7554/eLife.36317.007

**Figure supplement 5.** $\overline{DAF_i}$ as a function of various genomic predictors for the 1000G dataset.

DOI: https://doi.org/10.7554/eLife.36317.008

**Figure supplement 6.** $\overline{DAF_i}$ in 1000G populations as a function of recombination rate for various sites.

DOI: https://doi.org/10.7554/eLife.36317.009

**Figure supplement 7.** $\overline{DAF_i}$ as a function of recombination rate for different mutation types in 1000G individuals.

DOI: https://doi.org/10.7554/eLife.36317.010

**Figure supplement 8.** Influence of recombination on diversity (B-statistic).

DOI: https://doi.org/10.7554/eLife.36317.011

**Figure supplement 9.** $\overline{DAF_i}$ of WW + SS sites with RR $\geq$1.5 cM/Mb from the 1000G data set as a function of various covariates.

DOI: https://doi.org/10.7554/eLife.36317.012

**Figure supplement 10.** Genomic distribution of SNPs.

DOI: https://doi.org/10.7554/eLife.36317.013

As expected if purifying selection was removing deleterious variation predominantly in coding regions, we find a stronger effect of BGS in transcribed (TR) than in non-transcribed (NTR) regions, in the sense that $\overline{DAF_i}$ is more reduced in regions of low recombination in TR than in NTR regions (*Figure 1—figure supplement 6A*). At the same distance from exons (between ~0.001 and ~0.1 cM, *Figure 1—figure supplement 6B*), $\overline{DAF_i}$ is slightly larger for NTR than for TR regions suggesting that BGS is stronger in TR regions. However, $\overline{DAF_i}$ converges to similar values in high-recombination regions, in line with the view that BGS is not acting in these regions. Interestingly, BGS is clearly acting in NTR regions even when we focus on NTR regions more than 50 kb away from any transcribed region (*Figure 1B*). This result confirms that BGS is acting in NTRs (*Asthana et al., 2007*; *Comeron, 2014*; *Rands et al., 2014*), which could be either due to the presence of functional elements in these regions such as non-coding RNAs, histone marks, enhancers or insulators (*Kellis et al., 2014*; *Bonev and Cavalli, 2016*; *Van Nostrand et al., 2017*), or due to remote effects of exonic deleterious mutations on SNPs in NTR. However, since the influence of exonic regions on $\overline{DAF_i}$ is largely limited above 0.01 cM (*Figure 1—figure supplement 5B*), we suspect that functionally constrained elements are widespread in NTRs. Conservation scores have also been used to assess a potential effect of selection on $\overline{DAF_i}$. Sites associated to GERP RS scores between –2 and +2 are thought to be evolving neutrally in mammals (*Davydov et al., 2010*), but we still find a positive log-linear relationship between $\overline{DAF_i}$ and recombination rate for those sites (*Figure 1—figure supplement 5C*), suggesting that these sites are also influenced by BGS due to selection at linked sites. Note that we also find a positive relationship between $\overline{DAF_i}$ and recombination for more conserved sites that could be directly under negative selection (*Figure 1—figure supplement 5D–E*) suggesting that their diversity is also affected by BGS at neighbouring sites. These observations suggest that filtering by GERP score may not be sufficient to completely remove the effect of BGS. Since $\overline{DAF_i}$ patterns seem to be driven by BGS, we would expect that they are correlated with statistics that have been specifically developed to measure the extent of BGS in various regions of the genome, such as the *B*-statistic (*McVicker et al., 2009*). Indeed, the *B*-statistic measures the relative reduction in genetic diversity due to BGS and it ranges from 0 in regions highly affected by BGS to 1 in regions unaffected by BGS. As expected, we find that $\overline{DAF_i}$ and the average B-statistic, computed both in the same 20 recombination rate bins defined in *Figure 1A*, are highly correlated (*Figure 1—figure supplement 5F*). This result suggests that the average $\overline{DAF_i}$ and average *B*-statistic are affected by the same process, and thus that $\overline{DAF_i}$ provides information on the strength of background selection among a set of SNPs.

## Limits of BGS and evidence for biased gene conversion in regions of high recombination

Since the impact of BGS is mediated by recombination, BGS should have a minimal influence in regions of high recombination (*Hudson and Kaplan, 1995*; *Nordborg et al., 1996*). However, it has been shown that GC biased gene conversion (gBGC) is acting in GC/AT heterozygotes in these regions, particularly in the vicinity of recombination hotspots (*Spencer et al., 2006*), potentially increasing the frequency of G and C derived alleles (usually denoted as strong or S alleles, see *Lachance and Tishkoff, 2014*) as compared to A and T (denoted as weak or W alleles). We have thus examined the relationship between $\overline{DAF_i}$ and local recombination rate for three combinations of S and W alleles (*Figure 1C*, *Figure 1—figure supplement 6C*). If the ancestral allele is W and the derived allele is S (WS sites, *Figure 1C*, left), we see the same log-linear relation between $\overline{DAF_i}$ and recombination as if we consider all SNPs (*Figure 1A*). However, at SW sites (*Figure 1C*, center), $\overline{DAF_i}$ decreases for recombination rates above ~1.5 cM/Mb. This non-monotonic behavior at SW sites is consistent with gBGC favoring the transmission of G and C alleles, and thus decreasing the frequency of derived A and T alleles. Finally, for mutations not affected by gBGC (WW and SS sites), $\overline{DAF_i}$ increases with local recombination rate until it reaches a plateau starting at ~1.5 cM/Mb, which suggests that the effect of BGS is absent or strongly reduced above this recombination threshold (*Figure 1C*, right). This latter observation implies that the linear increase of $\overline{DAF_i}$ above 1.5 cM/Mb at WS sites (*Figure 1C*, left) is entirely due to gBGC. Note that the exact same pattern holds for SGDP populations (*Figure 1—figure supplement 4C*). Moreover, if we analyze all possible types of substitutions separately, gBGC appears to affect the 12 types of SNP according to whether the SNP

type belongs to the SW, WS, or WW +SS class (*Figure 1—figure supplement 7*. These results suggest that SNPs located in regions where recombination is higher than 1.5 cM/Mb are affected by gBGC and not by BGS (*Figure 1C*, *Figure 1—figure supplement 6C*). Therefore, WW and SS sites with a recombination rate above 1.5 cM/Mb (representing 2.88% and 2.94% of all SNPs for 1000G and SGDP datasets, respectively) should be optimal for demographic inference, as they appear to evolve mainly neutrally.

## BGS and gBGC affect the whole SFS

Since $\overline{DAF_i}$ increases with recombination rate (*Figure 1*), BGS does not simply amount to lowering the effective population size (*Charlesworth, 1994*; *Charlesworth et al., 1995*; *Hudson and Kaplan, 1995*), as this simple rescaling would not modify allele frequencies. BGS thus affects the SFS (*Zeng and Charlesworth, 2011*) in complex ways (*Nicolaisen and Desai, 2013*), and the comparison of sites that are differentially exposed to BGS allows us to better examine this influence. The SFS

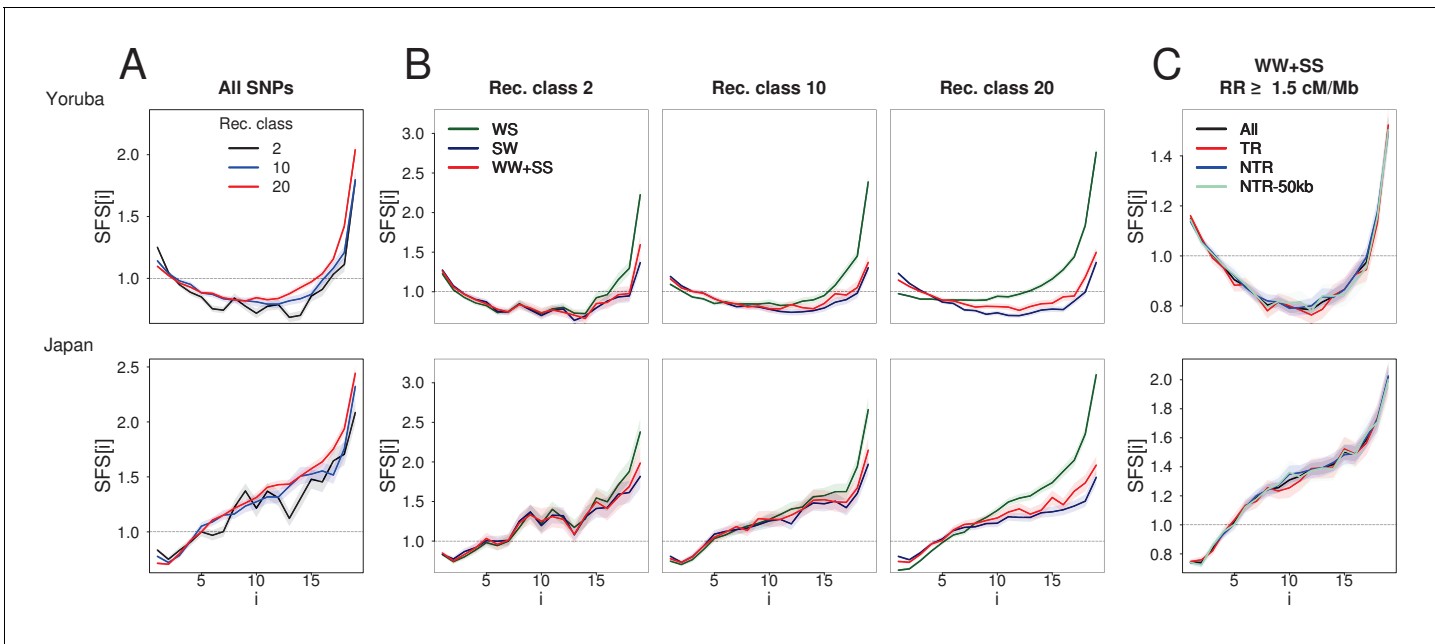

**Figure 2.** BGS and gBGC both have an impact on the SFS. Each panel corresponds to the normalized unfolded SFS of Yoruba (top, YRI) and Japanese (bottom, JPT) populations. (A) SFS computed for all SNPs in the 2nd, 10th and 20th recombination classes (as defined in *Figure 1*). For each panel, pairwise comparisons of the SFS are significant with p-values<$10^{-3}$ (see Materials and methods). The SFS for all ten 1000G populations are shown in *Figure 2—figure supplement 1*. (B) SFS for three gBGC mutation categories computed for three recombination classes. Note that WW and SS sites (in red) are unaffected by gBGC. All SFS are different from each other (site permutation test, p-values<$10^{-3}$) except for the Yoruba recombination class two between WS and WWSS where p=0.0135. (C) SFS for sites unaffected by BGS and gBGC (WW + SS sites with RR ≥1.5 cM/Mb). The four SFSs are not significantly different from each other at the 1% significance level, as revealed by a permutation approach (see Materials and methods). Shaded areas delimit 95% confidence intervals using a block-bootstrap strategy (see Materials and methods).
DOI: https://doi.org/10.7554/eLife.36317.014

The following figure supplements are available for figure 2:

**Figure supplement 1.** SFS of ten 1000G populations for sites belonging to three recombination classes.
DOI: https://doi.org/10.7554/eLife.36317.015

**Figure supplement 2.** (A) Normalized unfolded SFS of ten 1000G populations for sites belonging to three recombination classes in NTR regions more than 50 kb away from transcribed regions (NTR-50kb sites).
DOI: https://doi.org/10.7554/eLife.36317.016

**Figure supplement 3.** Impact of covariates on the normalized unfolded SFS for WW + SS sites with RR ≥ 1.5 cM/Mb (in red) on Yoruba (top) and Japan (bottom) population.
DOI: https://doi.org/10.7554/eLife.36317.017

**Figure supplement 4.** Impact of physical distance to phastCons conserved elements on the normalized SFS for WW + SS sites with RR ≥ 1.5 cM/Mb (in red) on Yoruba (left) and Japan (right) populations.
DOI: https://doi.org/10.7554/eLife.36317.018

computed in ten 1000G populations for different recombination classes (*Figure 2A*, *Figure 2—figure supplement 1*) shows distortions that are qualitatively similar in all populations, irrespective of differences in demographic history. As compared to the highest recombination class, the second-to-lowest recombination class (which is potentially the one most strongly affected by BGS) not only shows an excess of singletons, but also a deficit of intermediate and high frequency variants (*Figure 2A*). Similar distortions are also observed in non-transcribed regions, and even (but to a lower extent) in regions at least 50 kb away from transcribed regions (*Figure 2—figure supplement 2*), in line with our results for $\overline{DAF_i}$.

To understand the respective effects of gBGC and BGS on the SFS, we computed the SFS for subsets of mutations differentially affected by gBGC in the Yoruba (YRI) and Japanese (JPT) 1000G populations (*Figure 2B*). In line with previous work (*Lachance and Tishkoff, 2014*), we find that the difference between the SFSs of unbiased mutations (WW + SS) and biased mutations (SW and WS) increases with recombination rate. In particular, WS mutations show a deficit of low-frequency variants and an excess of intermediate- and high-frequency variants in regions of high recombination (*Figure 2B*). As previously recognized (*Katzman et al., 2011*; *Lachance and Tishkoff, 2014*), the excess of high-frequency variants at WS sites is not compensated by a corresponding deficit of high-frequency variants at SW sites, implying that gBGC could contribute to the increase of nearly fixed derived alleles that has previously been attributed to mislabelled ancestral states or positive selection (*Hernandez et al., 2007*).

## Impact on demographic inferences

To investigate the impact that the choice of SNPs may have on demographic inference, we estimated demographic parameters for the Yoruba and Japanese populations using three different SFSs (*Figure 3A*): the synonymous SFS commonly used in exome resequencing studies; the SFS inferred on non-transcribed regions at least 50 kb away from coding regions (NTR-50kb), and on our best-filtered dataset (WW + SS sites in ≥1.5 cM/Mb regions), hereafter called the 'neutral' dataset. Note that this neutral SFS was computed over both TR and NTR regions since they show the same SFS (*Figure 2C*). Interestingly, the SFS observed at synonymous sites differs markedly from that observed at neutral sites, as it comparatively shows a significant deficit of low-frequency variants and a large excess of high-frequency variants (*Figure 3A*, *Figure 3—figure supplement 1A*). It appears that this latter excess is due to gBGC, as it disappears when one computes the SFS on synonymous sites not affected by gBGC (*Figure 3—figure supplement 1B*).

Using a simple demographic model of a focal population going through three successive bottlenecks and receiving some migrants from surrounding populations (modelled as a ghost population for simplicity) (*Figure 3C*), we can fit almost perfectly the three SFSs (*Figure 3—figure supplement 2*). Yet, the inferred parameters differ considerably (*Supplementary file 3* - Table S3). For the Yoruba population, the differences in demography are especially important in the old periods (>100 ky, *Figure 3B*). With the neutral SFS, we nevertheless infer a more recent last bottleneck dated at the end of the Last Glacial Maximum (LGM), a more pronounced and more recent admixture event from surrounding populations. The ancient demography is markedly different with a significantly more ancient second bottleneck and a significantly lower ancient population size inferred from both synonymous and NTR-50 kb SFS. The Japanese demography inferred from the three data sets shows more similarity over the last 600 ky but the demography inferred from the neutral data set suggests a stronger recent bottleneck (pre LGM) and no population expansion as compared to what is inferred from the synonymous SFS neutral data set. Our results thus clearly show that very different demographies can be inferred from neutral and non-neutral SFSs. However, even though BGS and gBGC affect the SFS of populations with distinct histories in a qualitatively similar way, they have different consequences on their reconstructed demography. It thus appears difficult to predict how demographic parameters will be biased when using non-neutral SFS.

## Simulations of BGS reproduce observed patterns

To confirm that our observed patterns were compatible with background selection, we ran individual-based forward simulations implementing BGS with *SLiM* v. 2.3 (*Haller and Messer, 2017*) in populations having the demography estimated from neutral sites in the Japanese and the Yoruba populations (see *Supplementary file 3* - Table S3). Overall, the simulated BGS patterns qualitatively

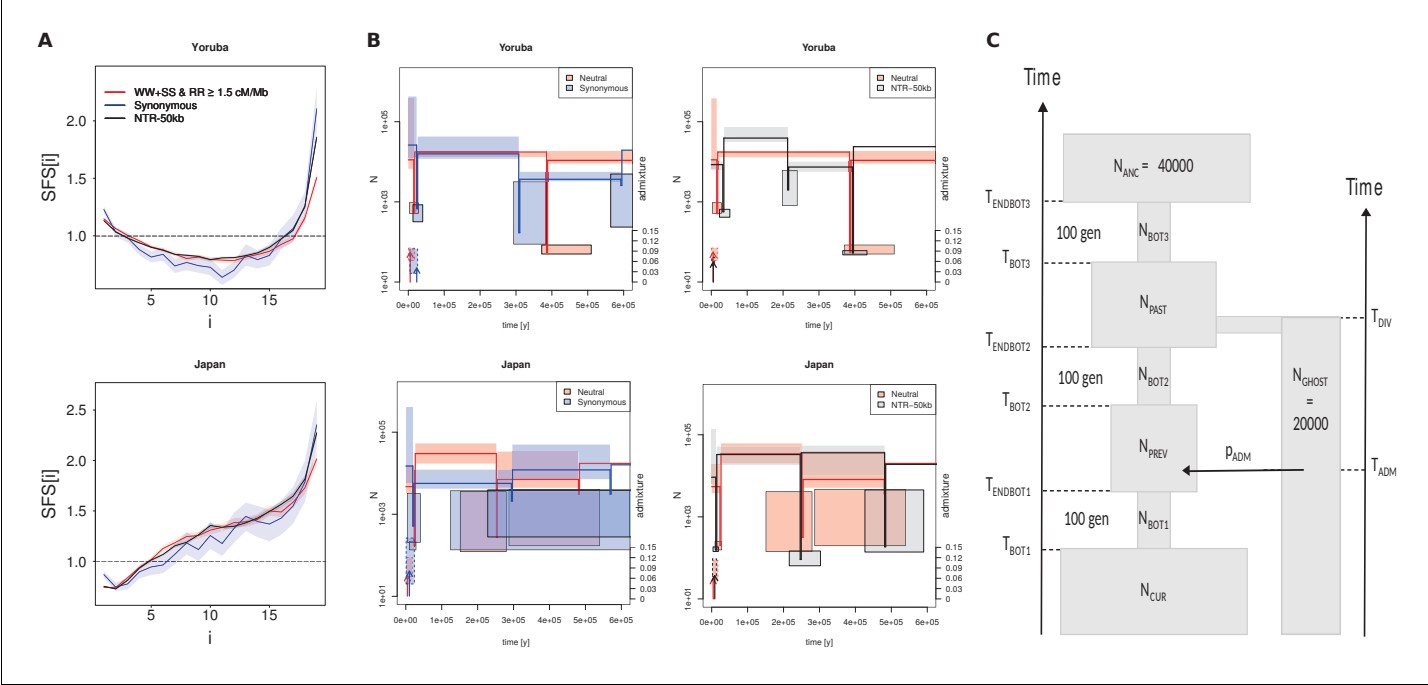

**Figure 3.** Recent demography inferred from three different datasets in the Yoruba and Japan 1000G populations. (A) Observed SFS computed on the three datasets. Neutral: neutral SFS computed on WW +SS sites in regions with recombination rate (RR) ≥1.5 cM/Mb; Synonymous: synonymous SFS; NTR-50kb: SFS computed on sites in non-transcribed regions more than 50 kb away from any transcribed region. Synonymous and NTR-50kb SFS are significantly different from neutral SFS with p-values<$10^{-3}$. (B) Comparison of inferred demographic events in the last 600,000 years under the model shown in panel C. Left and right panels compare neutral estimations to those of different datasets. All parameter values are given in *Supplementary file 3* - Table S3. Solid lines represent maximum-likelihood (ML) estimates of population sizes and bottleneck times. Vertical arrows indicate ML estimates of admixture times; their height is proportional to the admixture estimates shown on the right axis. Boxes delimit 99% confidence intervals obtained by a block-bootstrap approach (see Materials and methods). Boxes surrounded by a solid line are for bottleneck parameters (size and time), open boxes are for population size between bottleneck events, and boxes surrounded by a dashed line are for admixture rates and times. Note that bottlenecks have been modeled with a fixed duration of 100 generations, and the width of the boxes denotes the range covered by 99% of the bootstrap estimations. (C) Sketch of the demographic model used for SFS-based demographic inferences. The model includes three possible bottlenecks of a fixed duration of 100 generations in the direct ancestry of the sampled population, and it allows some sampled genes (a fraction $p_{ADM}$) to have ancestors coming from an unsampled (ghost) population at any time ($T_{ADM}$) since its divergence from the sampled population $T_{DIV}$ generations ago. Note that in this model, $N_{ANC}$ and $N_{GHOST}$ have been arbitrarily fixed to 40,000 and 20,000 (haploid sizes), respectively. Note also that the ghost population is used here to allow for some gene flow from some unspecified source, and so to account for the non-isolated nature of human populations.

DOI: https://doi.org/10.7554/eLife.36317.019

The following figure supplements are available for figure 3:

**Figure supplement 1.** Comparison of the unfolded SFS computed for synonymous, non-synonymous and 'neutral' sites (SS + WW sites with RR ≥ 1.5 cM/Mb) (A).

DOI: https://doi.org/10.7554/eLife.36317.020

**Figure supplement 2.** Observed and fitted SFS for the Yoruba and the Japanese samples under the demographic model shown in Fig.

DOI: https://doi.org/10.7554/eLife.36317.021

**Figure supplement 3.** Recent demography inferred from three different datasets in the Yoruba and Japan 1000G populations.

DOI: https://doi.org/10.7554/eLife.36317.022

match the observation very well (*Figure 4*, *Figure 4—figure supplement 1*, and *Figure 4—figure supplement 2*). As observed in real data (*Figure 1*), neutral sites simulated next to selected regions present a strong increase in $\overline{DAF_i}$ with recombination rate (*Figure 4A*), and the SFS at neutral sites shows a considerable excess of singletons and a deficit of intermediate- and high-frequency variants for low-recombination rates (*Figure 4B*), respectively. These results show that BGS can reproduce both the observed correlation between $\overline{DAF_i}$ and local recombination rates, and the observed distortions of the SFS in low-recombination regions.

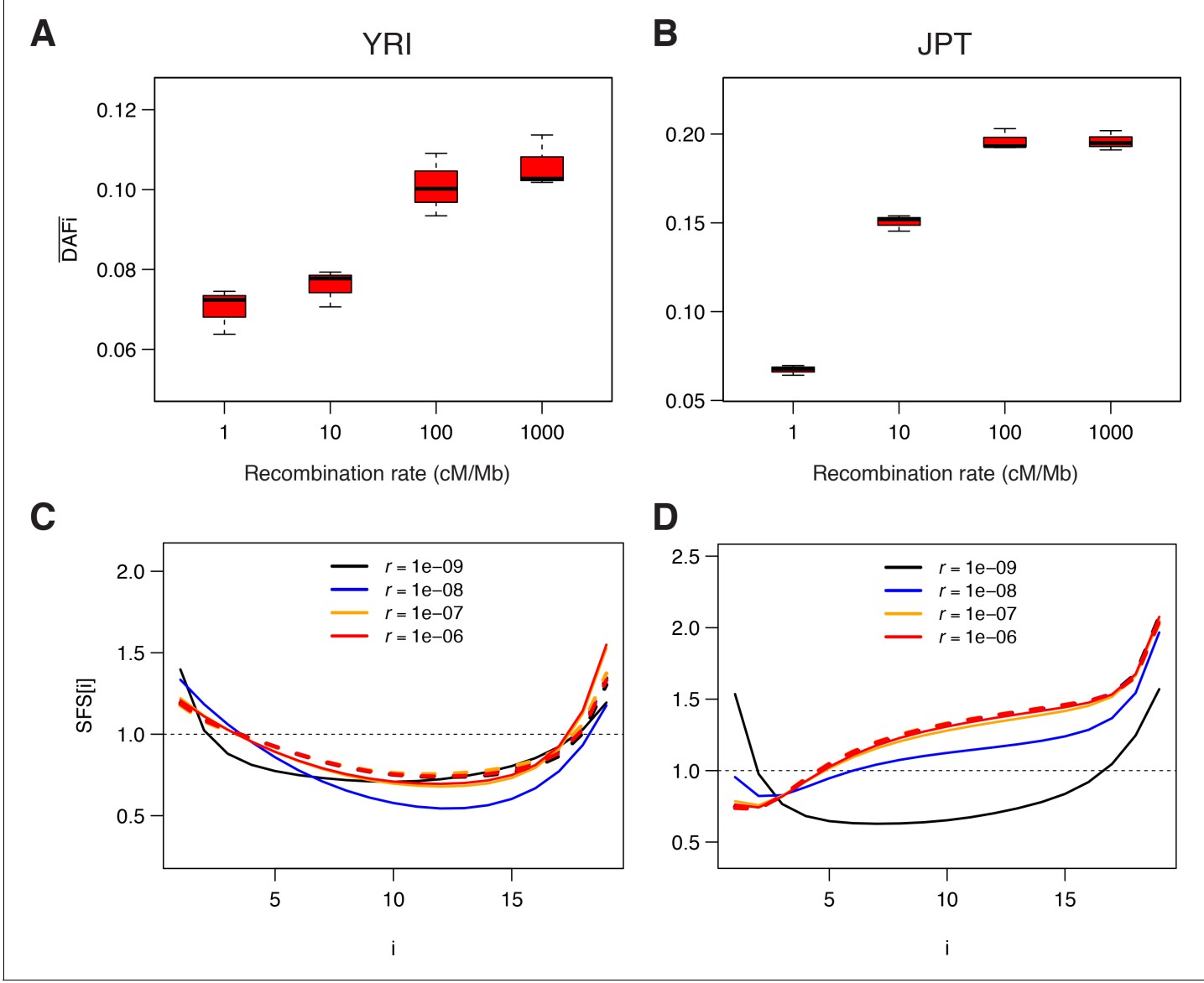

**Figure 4.** Genomic data simulated under a model of background selection (BGS). We used the demographic parameters estimated for the Yoruba (YRI) and Japanese (JPT) populations from neutral sites (WW + SS sites with RR $\geq$1.5 cM/Mb) as reported in *Supplementary file 3* - Table S3. Forward simulations of diploid individuals were performed with *SLiM* v. 2.3 (*Haller and Messer, 2017*). We simulated the evolution of a chromosome of 50 Mb made up of 1000 5 kb regions, each consisting of a 1 kb region experiencing purifying selections followed by a 4 kb region with neutral mutations. A, B) Average derived allele frequency per individual ($\overline{DAF_i}$). C, D) Unfolded normalized SFS. Solid and dashed lines correspond to simulations performed with and without BGS, respectively. The transition to effective neutrality occurs between a recombination rate of 1e–8 (blue curve) and 1e–7 (orange curve), a range that includes our proposed threshold of 1.5 cM/Mb.
DOI: https://doi.org/10.7554/eLife.36317.023

The following figure supplements are available for figure 4:

**Figure supplement 1.** Simulated SFS under neutrality and with background selection for various recombination rates.
DOI: https://doi.org/10.7554/eLife.36317.024

**Figure supplement 2.** Simulated $\overline{DAF_i}$ and SFS under neutrality and with background selection for various recombination rates.
DOI: https://doi.org/10.7554/eLife.36317.025

# Discussion

Delineating the neutrally evolving part of the human genome remains a challenge, as variation in the intensity of recombination, mutation, and selection are increasingly recognised as having a strong effect on observable genomic diversity in humans (*Corbett-Detig et al., 2015*; *Elyashiv et al., 2016*) and other organisms (e.g. *Elyashiv et al., 2016*; *Ravinet et al., 2017*). Here, we have shown that a surprisingly large proportion (up to 95%) of our genome might be affected by background selection (BGS) and/or GC-biased gene conversion (gBGC). These two processes, which both depend on recombination, strongly affect observed measures of genetic diversity along the genome and can lead to biased demographic inference if not properly taken into account (*Figure 3*).

We have interpreted the striking linear relationship observed between $\overline{DAF_i}$ and recombination rate (*Figure 1*) as evidence for the pervasive effect of BGS but other processes could in principle lead to a similar relationship. For instance, a mutagenic effect of recombination could lead to an increased diversity in regions of high recombination (*Hellmann et al., 2003*). The examination of extremely low-frequency mutations, which should be enriched for new mutations, did not reveal any association between recombination rate and the density of new mutations in a large human sample (*Schaibley et al., 2013*), but a more recent study of de novo mutations suggested the existence of such a correlation (*Francioli et al., 2015*). Alternatively, a correlation between mutation and recombination rates could occur if these rates were both affected by the same process, such as replication timing (*Stamatoyannopoulos et al., 2009*; *Koren et al., 2012*) or transcription rate (*Gerton et al., 2000*; *Park et al., 2012*). However, a mere correlation between mutation and recombination rates cannot explain two key aspects of our observations. First, $\overline{DAF_i}$ plateau at high recombination rates once the effect of gBGC is removed (*Figure 1C*), whereas it should continue increasing if only mutation-recombination correlation was driving the relationship between $\overline{DAF_i}$ and recombination. Second, we find a significant difference in the shape of SFS computed in regions of low and high recombination (*Figure 2A*), even though mutation rate should have no effect on the shape of the SFS. To better investigate the effect of a possible mutation-recombination correlation, we have used the fact that $\overline{DAF_i}$ is correlated with the B-statistic (*Figure 1—figure supplement 5F*), for which a simple model (*Hudson and Kaplan, 1995*) predicts its value as a function of mutation and recombination rates. We find that the B-statistics inferred by *McVicker et al. (2009)* are significantly better fitted as a function of the recombination rate if we assume a log-log linear relationship between recombination and deleterious mutation rate than if we impose a constant mutation rate across the genome (*Figure 1—figure supplement 8*). Interestingly, under our log-log linear model, both the observed and predicted B-statistics reach a plateau value of ~0.9 above a recombination rate of ~1.5 cM/Mb. This pattern remains if we only consider subsets of SNPs (e.g. WW + SS sites; *Supplementary file 4* - Table S4). Therefore, these results suggest that in addition to BGS and gBGC, some correlation between mutation and recombination rate is required to best explain our observed patterns. Moreover, given the relationship observed between B-statistics and $\overline{DAF_i}$ (*Figure 1—figure supplement 5F*), the reduced effect of recombination on B above 1.5 cM/Mb should translate into a similar absence of change in $\overline{DAF_i}$ above the same threshold, thus explaining the plateau we see in *Figure 1C* above 1.5 cM/Mb.

The occurrence of pervasive positive selection, either in the form of soft or hard sweeps (*Kern and Hahn, 2018*) or of positive selection on polygenic traits (*Boyle et al., 2017*) in our genome could also lead to a correlation between genetic diversity and recombination, as the effect of selection on linked neutral sites should decrease with recombination. However, positive selection should lead to an increase of both low- and high-frequency variants in the SFS (*Fay et al., 2000*; *Hernandez et al., 2007*; *Huber et al., 2016*; *Pavlidis and Alachiotis, 2017*), whereas we only observe an increase of low-frequency variants in low-recombination regions where the effect of selection should be strongest (*Figure 2A*), which is the expected effect of BGS (*Figure 4*).

The exact proportion of the genome that is influenced by selection is still the source of an intense debate (*Bernstein et al., 2012*; *Rands et al., 2014*; *Graur, 2017*; *Kern and Hahn, 2018*). Here, we show that up to 80–85% of the human genome is probably affected by background selection (BGS), an effect that is not subtle (*Reed et al., 2005*) and that is visible from single individuals genomes (*Figure 1—figure supplement 5A*). Even though our estimate of the fraction of the human genome influenced by BGS matches relatively well with that reported to be biochemically functional by the ENCODE consortium (*Bernstein et al., 2012*), our results do not imply that 80–85% of the human

genome is functional. They rather show that functional sites that are the direct target of purifying selection in both coding and non-coding regions (potentially representing 8–15% of the genome, *Rands et al. (2014)*; *Graur, 2017*) have an important but indirect influence on most of the genome.

As expected, the effect of BGS is clearly mediated by local recombination rate, but it extends well beyond coding regions in humans (*Hernandez et al., 2011*) (*Figure 1*), and it is thus not restricted to species with a large effective size (*Corbett-Detig et al., 2015*). Our results also show that the influence of gBGC is not restricted to recombination hotspots (*Spencer et al., 2006*; *Glémin et al., 2015*), but that it has also a strong footprint in regions with a recombination rate larger than 1.5 cM/Mb, but note that it could affect (to a lesser degree) regions with an even lower recombination rate (see *Figure 1—figure supplement 5D*. These regions represent about 15.9% and 16.2% of the polymorphic positions for the 1000G and SGDP datasets, respectively. Taken together, BGS and gBGC thus affect more than 95% of the polymorphic sites in our genome, and we have identified only a small fraction of all genomic SNPs (~3%, *Supplementary file 2* - Table S2) that can be considered as evolving neutrally.

Interestingly, our neutral SNPs are found in both transcribed and non-transcribed-regions (*Figure 2C*), and they are enriched close to telomeric regions (*Figure 1—figure supplement 10*), where BGS is predicted to be weaker (*Charlesworth, 2012*). Whereas SNPs included in our best-filtered set are evolving mostly neutrally, it does not imply that all other SNPs are influenced by BGS and gBGC. Indeed, our way of identifying selection and biased gene conversion is indirect and operates on arbitrarily defined recombination-rate categories. Thus, $\overline{DAF_i}$ cannot be used to identify the presence of selection at the SNP level or in small genomic regions, or inversely, the presence of neutral SNPs in low recombining segments between recombination hotspots. A more precise mapping of selected genomic segments could use information on the positions of known functional elements (*Siepel et al., 2005*; *Kellis et al., 2014*; *Rands et al., 2014*; *Elkon and Agami, 2017*) or B-statistics (*McVicker et al., 2009*; *Elyashiv et al., 2016*), which could also be used to evidence neutrally evolving regions in both low- and high-recombination regions.

To investigate if and how $\overline{DAF_i}$ depends on potential co-variates within our neutral set of SNPs, we have examined its relationship with several statistics, such as B-statistics or the distance (in map units) to exons, as well as distances to conserved elements and to recombination hotspots. In our neutral set, we find virtually no relationship between $\overline{DAF_i}$ and recombination rate, with average $\overline{DAF_i}$ remaining close to its mean value of 0.146 (*Figure 1—figure supplement 9A*), but we find a negative relation with the distance to recombination hotspots (*Figure 1—figure supplement 9B*, a positive relationship with distance to conserved elements and with B-statistics (*Figure 1—figure supplement 9C–D*), and a small positive correlation with distance to exons ($\overline{DAF_i}$ varies from 0.145 to 0.15, close to the average, *Figure 1—figure supplement 9E*). It thus seems that recombination hotspots still play a role in decoupling selected from neutral sites, and that sites furthest away from hotspots might still be slightly sensitive to BGS. Purifying selection in phastCons conserved elements (*Siepel et al., 2005*) is also exerting a strong negative pressure on derived allele frequencies, with average $\overline{DAF_i}$ below 0.14 at sites less than 0.0003 cM away from these elements (which correponds approximatively to a distance of 200 bp if RR = 1.5 cM/Mb). Contrastingly, being further than 0.05 cM away from these conserved elements allows $\overline{DAF_i}$ to rise above 0.16, an average value that is barely reached for sites with associated mean *B* values close to 1. These results suggest that phastCons elements represent the covariate that has the strongest remaining influence on $\overline{DAF_i}$ within our neutral set.

The SFS of each population is affected by BGS and gBGC (*Figure 2*, *Figure 2—figure supplement 1*), and the demography inferred from neutrally evolving SNPs differs markedly from that based on synonymous sites or sites in non-transcribed regions (*Figure 3A*). However, we show that BGS and gBGC can have different impacts on the inferred demography of the populations. For instance, we found that they lead to an underestimation of the age of a bottleneck and an overestimation of the magnitude of a demographic expansion in the Yoruba population, but we do not observe such strong biases in the Japanese population. It therefore appears difficult to predict the specific biases introduced by these evolutionary forces on demographic inference, except perhaps under simple evolutionary scenarios (*Ewing and Jensen, 2016*). We therefore suggest that future studies of demographic history should be based on a set of markers that is minimally influenced by these non-neutral forces.

We have also computed the observed SFS for subsets of neutral SNPs with various values of the covariates mentioned above (*Figure 1—figure supplement 9*). SNPs in the 1st and 4th distance-quartiles to hotspot show similar SFS, with a slight excess of singletons and high-frequency variants for the sites furthest to hotspots (*Figure 2—figure supplement 3A*. Even though conserved elements had the strongest influence on $\overline{DAF_i}$, the SFSs computed at sites belonging to the 4th distance quartile and to all sites still look very similar, especially in the Japanese population, while sites in the 1st distance quartileshow an excess of singletons and a deficit of high-frequency variants (*Figure 2—figure supplement 3B*. Exonic and non-exonic SFSs within our neutral set differ mainly by increased frequencies of singletons for exonic SNPs, yet the removal of exonic SNPs has no impact on the SFS (*Figure 2—figure supplement 3C*). In conclusion, even though exonic SNPs and those located close ($\leq$0.0003 cM) to phastCons elements show different SFS shapes (*Figure 2—figure supplement 4*), their removal from our neutral set would have no major effect on the shape of the SFS, since they represent only a small fraction (2.2% and 16.9% respectively) of the SNPs in our neutral set.

It is interesting to compare our neutral set of SNPs to another previously defined set of neutral regions of the human genome that has been used as a reference for demographic inferences in a series of studies (e.g. *Gronau et al., 2011*; *McManus et al., 2015*; *King and Wakeley, 2016*; *Veeramah et al., 2018*). *Gronau et al. (2011)* have identified a set of 37,574 potentially neutral regions of 1 kb in length with carefully chosen properties (e.g. at least 1 kb away from exons and 100 bp away from phastCons elements, without CpG sites, separated by at least 50 kb, without recombination hotspots). The SFS computed on this alternative neutral set departs significantly from our neutral set, with a significant excess of singletons, a deficit of sites with intermediate allele frequencies, and an excess of nearly fixed variants, a pattern that can be explained by the action of both BGS and gBGC (*Figure 3—figure supplement 3A*. Since a large B-statistic is also indicative of relaxed BGS, one could be tempted to use regions associated with *B* values larger than 0.9 as being potentially neutral. However, we see that its SFS also departs from that of our neutral set, with a small deficit of singleton and an excess of other frequency classes in Yoruba, and a slight excess of high-frequency variants in Japan (*Figure 3—figure supplement 3A*). These differences in SFS shapes also lead to inferred demographies that are markedly different from that inferred from our own neutral set, and this especially for the Yoruba population (*Figure 3—figure supplement 3B*). We suspect that the main discrepancy with our neutral set is the presence of gBGC in regions with *B* > 0.9, such that filtering out SW and WS SNPs may result in a good alternative data set on which to perform demographic inferences

Methods of demographic inference based on whole genomes (e.g. *Li and Durbin, 2009*; *Sheehan et al., 2013*; *Schiffels and Durbin, 2014*) should also be sensitive to BGS and gBGC, since they assume that heterozygosity levels within individuals is not driven by local recombination rates nor selection. In this respect, the history of human populations as well as that of other species might be more readily inferred from methods that can conveniently analyze restricted sets of neutrally evolving sites interspersed across the genome. Similarly, other types of inference using a biased neutral SFS as a reference could also be affected, such as inferences of the distributions of fitness effects (DFE) (*Keightley and Eyre-Walker, 2010*; *Kim et al., 2017*; *Tataru et al., 2017*), even though the magnitude of the effect remains to be investigated. In conclusion, we show that BGS and gBGC had a pervasive effect on most of our genome, but that we can conveniently define a set of sites (representing about 3% of all polymorphic sites of both 1000G and SGDP datasets) that should not be too influenced by these two evolutionary forces, even though some sites close to conserved elements could still be affected by BGS. Contrary to previously used sets of SNPs, these sites should lead to essentially unbiased demographic inferences and serve as a reference for future demographic reconstructions in humans. Due to its simplicity, our approach can be readily applied to any species for which a recombination map is available.

## Materials and methods

### Datasets

We analyzed two distinct whole genome datasets. The first one consisted of 100 individuals from ten 1000G populations (*Auton et al., 2015*). For each 1000G population, we selected the ten

individuals with the highest depth of coverage (coverage >10×), such as to maximize the number of sites having no missing data. We also analyzed 20 individuals from panel C of the Simons Genome Diversity Project (SGDP) (*Mallick et al., 2016*). These individuals were selected from ten SGDP populations that were geographically close to those analyzed for the 1000G project. Coverage was higher for the SGDP individual and ranged between 31 × and 64× (see *Supplementary file 1* - Table S1 for IDs and location of the 1000G and SGDP samples).

## Data processing and annotations

We processed the 1000G and SGDP datasets identically. We removed all sites with any missing data and kept only diallelic sites from autosomal chromosomes. The ancestral state of each variant in these genomes was set to the chimpanzee reference genome (panTro4 genome assembly) to avoid any discrepancy between African and non-African populations. Only diallelic SNPs for which one of the variants observed in the 1000G or SGDP datasets corresponded to the chimpanzee ancestral state were kept for later analyses. In addition, we removed the CpG sites that present a peculiar mutation profile and are correlated with recombination rate (*Arbeithuber et al., 2015*). We used the LD-based Yoruba-specific recombination map from the 1000 Genomes project (*Frazer et al., 2007*) to obtain the local recombination rate (RR) surrounding each SNP. We also estimated local RR by using three other maps: the LD-based CEU or JPT-specific recombination maps (*Frazer et al., 2007*) and the sex-averaged pedigree-inferred deCode map (*Kong et al., 2010*). For each of these maps, we filtered out SNPs without RR information (see *Supplementary file 2* - Table S2). We used the Yoruba-specific map to define hotspots as regions with RR >10 cM/Mb. Using Biomart (http://grch37.ensembl.org/biomart/martview/), we assigned SNPs to transcribed (TR) and non-transcribed regions (NTR). For each site, we inferred the distance to the closest exonic region in cM and in bp using the Ensembl exon positions (ftp://ftp.ensembl.org/pub/grch37/release-90/gtf/homo_sapiens/Homo_sapiens.GRCh37.87.gtf.gz). The B-statistic (*McVicker et al., 2009*) (indicative of the strength of local background selection) associated with each SNPs was retrieved from http://www.phrap.org/othersoftware.html and lifted over from the hg18 to the hg19 reference genome using the UCSC liftOver tool. Genomic Evolutionary Rate Profiling (GERP) rejection scores (*Davydov et al., 2010*) that quantify the level of evolutionary constraint acting on polymorphic sites and conserved elements identified using PhastCons on the primate subset of 46 vertebrates (*Siepel et al., 2005*) were downloaded from the UCSC platform (*Speir et al., 2016*). The number of SNPs from the 1000G and SGDP datasets retained for each filter is reported in *Supplementary file 2* - Table S2. We finally retrieved 37,574 potentially neutral regions of 1 kb (e.g. *Gronau et al., 2011*; *McManus et al., 2015*; *King and Wakeley, 2016*; *Veeramah et al., 2018*) from http://compgen.cshl.edu/GPhoCS/data.php to make comparisons between our neutral set of SNPs to another possible sets.

## Estimating the impact of GC-biased gene conversion (gBGC)

As gBGC favors strong (abbreviated as S, and representing C and G bases) compared to weak (abbreviated as W, and representing A and T bases) alleles, we defined three groups of SNPs according to the expected consequences of gBGC: (1) SNPs for which the derived state is favoured (WS sites); (2) SNPs for which the ancestral state is favoured (SW sites), and (3) SNPs on which gBGC has no effect (WW or SS sites).

## Average derived allele frequency per individual ($\overline{DAF_i}$)

To quantify a local effect of selection and/or gBGC, we used the average derived allele frequency per individual ($\overline{DAF_i}$), where this average is computed over a set of sites found polymorphic in a collection of individuals. We show in the following that this statistic is ideally suited to evidence the potential effect of selection (or mutation), as difference in the demography of the populations from which individuals are sampled should not translate into different values of this statistic among individuals.

Start by considering a single non-recombining locus ($k$) with mutation rate $u_k$, and for the sake of simplicity, let us consider just two individuals $i$ and $j$, drawn from two different populations. Note that the same reasoning can be extended to an arbitrary number of individuals drawn from an arbitrary number of populations. Now, suppose that the two homologous alleles of these individuals have coalesced $t_i$ and $t_j$ generations ago, and that the most recent common ancestor of these four

homologous alleles is $t_{\text{global}}$. Now, the frequency of the derived allele in individual $i$ at the $k$-th locus is simply given by

$$DAF_{ik} = \frac{n_{ik}}{2S_{tot,k}} \qquad (1)$$

where $S_{tot,k}$ is the total number of sites that are polymorphic at this $k$-th locus for this sample of two individuals, and $n_{ik}$ is the number of derived alleles observed in individual $i$. Since $n_{ik}$ is the number of heterozygous sites ($Het_{ik}$) plus two times the number of homozygous derived sites ($Hom_{ik}$) (see *Figure 1—figure supplement 1*), the expected value of $n_{ik}$ can be expressed as a function of $t_{\text{global}}$ and the mutation rate $u_k$ as

$$E(n_{ik}) = E(Het_{ik} + 2Hom_{ik}) = 2u_k t_{ik} + 2u_k(t_{global,k} - t_{ik}) = 2u_k t_{global,k}, \qquad (2)$$

which does not depend on $t_{ik}$, the coalescence times between homologous alleles in individuals 1 or 2, as illustrated in *Figure 1—figure supplement 1*. Therefore, $E(n_{jk}) = E(n_{ik}) = 2u_k t_{global,k}$, and

$$E(DAF_{ik}) = E(n_{ik})/E(2S_{tot,k}) = t_{global,k}/T_{tot,k}, \forall i, \qquad (3)$$

where $T_{tot,k}$ is the total tree length at the $k$-th locus. Since the average derived allele frequency computed over an arbitrary number of unlinked loci $m$ is obtained as the ratio of the total number of derived alleles over twice the total number of polymorphic sites, its expectation is then obtained as

$$E(\overline{DAF_i}) = \frac{E(n_i)}{E(2S_{tot})} = \frac{\sum_k^m E(n_{ik})}{\sum_k^m E(2S_{tot,k})} = \frac{\sum_k^m u_k t_{global,k}}{\sum_k^m u_k T_{tot,k}}, \qquad (4)$$

an equation that is valid irrespective of the number of individuals and populations sampled if one computes the number of derived alleles over all sites found polymorphic in the collection of individuals. If the mutation rate is uniform across loci, then equation (1.4) simplifies to

$$E(\overline{DAF_i}) = \bar{t}_{global}/\bar{T}_{tot}, \qquad (5)$$

which only depends on the average global coalescence time of the total sample $\bar{t}_{global}$, and on the average tree length over all loci $\bar{T}_{tot}$, and not on the coalescence times in each population. Therefore, even though $E(\overline{DAF_i})$ depends on the overall demography of the collection of individuals and on the composition of the samples, which both condition the global $t_{\text{MRCA}}$ and total tree lengths, the specific demographic histories of the sampled populations will not translate, in expectation, into different $\overline{DAF_i}$ among individuals examined for the same set of loci. Selection in some portion of the genome will affect $t_{\text{MRCAs}}$, which should thus translate into differences in $\overline{DAF_i}$ computed for these regions. Differences in mutation rates across the genome might also affect $\overline{DAF_i}$ for some regions, but should not lead to individual differences, unless mutation rates are different in specific populations.

For both SGDP and 1000G data sets, we ranked SNPs according to their associated recombination rate and binned them into 20 equal-sized classes of increasing recombination rates. We performed a similar binning for the different groups of SNPs we considered (the three types of mutations, within a transcribed region or not, etc.) or after ranking SNPs according to their distance to the nearest exon, to hotspots or to conserved elements. We then computed $\overline{DAF_i}$ for each bin $b$ as $\overline{DAF_{ib}} = n_{ib}/(2S_{tot,b})$.

## Site frequency spectrum

We estimated the unfolded SFS for ten 1000G population samples using different filters (e.g. different recombination classes, different types of mutations). The SFS was then normalized (*Lapierre et al., 2017*) by dividing each entry by its expectation in a stationary population. To estimate if two SFSs are statistically different, we used a permutation approach. We first computed a distance between the two SFS as the sum of the squared difference in site frequencies over all SFS entries (noted $D_{obs}$). We divided the SNPs into three categories: those shared by the two SFS (if any), and those that were private to one of the SFS. We then randomly permuted sites among the two latest categories and re-evaluated the distance noted $D_{est}$. When one SFS was based on a

subset of variants from another SFS, we subsampled sites from the largest dataset and re-evaluated $D_{est}$. We repeated the permutations or the resampling procedure 1000 times and estimated a $p$ value as the frequency of $D_{obs} \geq D_{est}$.

## Block-bootstrap procedure

For each filter (e.g. per recombination class or per type of mutation), we identified sets of 100 adjacent SNPs along the genome and we sampled them with replacement such as to keep the same number of sites as in the non-bootstrapped set when computing statistics of interest ($\overline{DAF_i}$, SFS). We repeated the sampling 1000 times to obtain 1000 block-bootstrap sets of SNPs. 95% confidence intervals were computed by identifying the 2.5 and 97.5 quantiles of the resulting bootstrap distributions.

## Demographic inference

We estimated the parameters of the demographic model shown in *Figure 3C* from the SFS of two 1000G populations (Japan and Yoruba) using the program *fastsimcoal2* (*Excoffier et al., 2013*) ver 2.6. We used the following command line options:

./fsc26 -t pop.tpl -n200000 -d -e pop.est -M -l25 -L50 -q −0 -C1 -c1 -B1,

where *pop* denotes either the Japan or the Yoruba population. We used the *tpl* and *est* setting files defined in Supplementary file SF1. For each population, we performed 50 independent estimations and retrained the parameters that maximized the model likelihood. The confidence intervals of the parameters were estimated from 100 block-bootstrapped SFS obtained in a way similar to that described above. For each population, estimations were performed on each bootstrap dataset independently, using the maximum likelihood (ML) parameters values estimated above as initial values. Since we started parameter estimation close to the observed ML values, we only did five estimations per bootstrap and retained the parameters with maximum associated likelihood. A 99% confidence interval was then obtained for each parameter by estimating the 0.5% and 99.5% quantiles of its resulting bootstrap distribution.

## Individual-based simulations

We performed individual-based simulations using the software SLiM v. 2.3 (*Haller and Messer, 2017*) to check that BGS could reproduce observations. We simulated the demographic scenario inferred from the 'neutral' SFS (i.e. from WW + SS sites with $r \geq 1.5$ cM/Mb) for the Japanese (JPT) and Yoruba (YRI) 1000G populations as described above (Demographic inference). We simulated a linear genome of 50 Mb made up of 1000 regions of 5 kb. Each of these regions consisted of a 1 kb stretch experiencing purifying selection against deleterious mutations, followed by a 4 kb stretch with neutral mutations. We also simulated an alternative genomic architecture with 10,000 regions of 500 bp, each consisting of a 100 bp stretch under purifying selection, followed by a 400 bp stretch with neutral mutations. For computational efficiency, we scaled the inferred event times and population sizes by a factor of 0.1 and give below the rescaled values. We set the per-site mutation rate to $1.25 \times 10^{-7}$ for deleterious and neutral mutations. The fitness contribution of all deleterious mutations was $1 - s$ in homozygous form and $1 - s/2$ in heterozygous form. The fitness of individuals was computed multiplicatively across sites. We ran independent simulations for four recombination rates ($r = 10^{-9}$, $10^{-8}$, $10^{-7}$, and $10^{-6}$). For each demographic scenario and recombination rate, we simulated a scenario with background selection ($s = -0.1$) and a neutral scenario ($s = 0$). For each parameter combination, we performed 100 independent replicates starting with a period of $4 \times N_{ANC}$ generations, where $N_{ANC}$ is the number of haploid genomes in the ancestral population (*Figure 4—figure supplement 1*). We set $N_{ANC} = 4000$ for both the Yoruba and Japanese simulation. At the end of each simulation, we output the full population and computed the number of derived alleles for each individual across a fixed number arbitrarily set to 40,000 SNPs, subsampled from all SNPs. These 40,000 SNPs were subsampled individually for each replicate simulation. The SFS of the population was subsampled to 10 individuals (i.e. 20 haploid genomes) following

*Nielsen et al. (2005)* as $p_{i,20} = k^{-1} \sum_{j=1}^{k} \binom{f_j}{i} \binom{n_j - f_j}{20 - i} / \binom{n_j}{20}$, where $k$ is the total number of SNPs in the dataset, and $n_j$ and $f_j$ are the number of haploid genomes in the full sample and the number of derived alleles in the full sample at the *j*th SNP, respectively (see also *Liu et al., 2017*). We

computed the SFS separately for each replicate simulation, and then calculated the mean and the 2.5 and 97.5 percentiles across these replicates for each entry $p_{i,20}$. We normalized the SFS as described above (subsection SFS).

## Accounting for a correlation between mutation and recombination rates

To model a potential correlation between mutation and recombination, we assumed that the per-base pair deleterious mutation rate $u_d$ depends on the local recombination rate $r$ as

$$u_d(r) = u_0 r^b.$$

This assumption implies a log-log linear relationship between mutation and recombination, with an intercept of $log(u_0)$ and a slope of $b$. In the special case of $b = 0$, mutation is independent of recombination. We then modified the approximate BGS model of *Hudson and Kaplan (1995)* by substituting $u_d(r)$ for the deleterious mutation rate. The reduction in the nucleotide diversity a at a focal site due to BGS is then predicted to be

$$B = \frac{\pi}{\pi_0} \approx exp\left(-\frac{u_d(r)}{r}\right) = exp\left(-\frac{u_0 r^b}{r}\right) = exp\left(-u_0 r^{(b-1)}\right) \tag{6}$$

where $\pi_0$ is the baseline nucleotide diversity in the absence of BGS, and $\pi$ is the effective nucleotide diversity with BGS. We fit this modified BGS model to the relationship between the *B*-statistic from *McVicker et al. (2009)* and the recombination rate associated with our polymorphic SNPs using the method of non-linear least squares as implemented in the nls function in R v 3.4.4 (*R core Team, 2018*). We then used the Akaike information criterion (AIC, *Akaike, 1974*) to compare this extended BGS model to the original BGS model in which the mutation rate does not depend on the recombination rate ($b = 0$). Note that *McVicker et al. (2009)* obtained their *B*-statistics by fitting a more complex BGS model to polymorphism and recombination data (assuming no specific correlation between recombination and mutation). However, the model of *Hudson and Kaplan (1995)* used here is just a simplified version of that used by *McVicker et al. (2009)*. It assumes that neutral sites on which diversity is measured are in the middle of a region containing sites under negative selection, that recombination rates are uniform in the considered region, and that selection coefficients at deleterious sites are small relative to the total recombination rate in the region. These assumptions seem reasonable except for sites that are very close to recombination hotspots or close to telomeres, but we expect a qualitatively global agreement between these two models. An exact quantitative match is not required here, since our goal here is simply to assess whether a correlation between mutation and recombination rates needs to be invoked rather than to accurately estimate the parameters of the model ($u_0$ and $b$).

## Acknowledgements

We thank Anthanasios Kousathanas, Guillaume Achaz, Etienne Patin, Lluis Quintana-Murci, Sylvain Glémin, and Rasmus Nielsen for informative discussions on the subject, and Montgomery Slatkin for his careful reading and helpful comments on the manuscript. FP and SA have been supported by a Swiss NSF grant No 310030B-166605 to LE. LE was also supported by the Institut Pasteur in Paris and by a Visiting Miller Professorship grant from the University of Berkeley during his sabbatical. Calculations were performed on the UBELIX (http://www.id.unibe.ch/hpc) cluster of the University of Bern. The source code and setting parameters to reproduce the analyses are available at http://data-dryad.org/review?doi=doi:10.5061/dryad.t76fk80.

# Additional information

## Funding

| Funder | Grant reference number | Author |
|---|---|---|
| Schweizerischer Nationalfonds zur Förderung der Wissenschaftlichen Forschung | 310030B-166605 | Laurent Excoffier |
| University of California Berkeley | Visiting Miller Professorship | Laurent Excoffier |

The funders had no role in study design, data collection and interpretation, or the decision to submit the work for publication.

## Author contributions

Fanny Pouyet, Simon Aeschbacher, Conceptualization, Formal analysis, Investigation, Methodology, Writing—original draft, Writing—review and editing; Alexandre Thiéry, Data curation; Laurent Excoffier, Conceptualization, Resources, Formal analysis, Supervision, Funding acquisition, Validation, Investigation, Methodology, Writing—original draft, Project administration, Writing—review and editing

## Author ORCIDs

Fanny Pouyet (iD) http://orcid.org/0000-0001-5614-6998
Laurent Excoffier (iD) http://orcid.org/0000-0002-7507-6494

## Decision letter and Author response

Decision letter https://doi.org/10.7554/eLife.36317.038
Author response https://doi.org/10.7554/eLife.36317.039

# Additional files
## Supplementary files

• Supplementary file 1. Table S1: Individuals analyzed in this study
DOI: https://doi.org/10.7554/eLife.36317.026

• Supplementary file 2. Table S2: Properties of the different data sets after filtering
DOI: https://doi.org/10.7554/eLife.36317.027

• Supplementary file 3. Table S3: Maximum-likelihood estimates of parameters of the demographic model shown in *Figure 3—figure supplement 3*, with limits of 99% confidence intervals.
DOI: https://doi.org/10.7554/eLife.36317.028

• Supplementary file 4. Table S4: Parameter estimates and quality-of-fit statistics for models of BGS with and without a mutation–recombination correlation.
DOI: https://doi.org/10.7554/eLife.36317.029

• Transparent reporting form
DOI: https://doi.org/10.7554/eLife.36317.030

## Data availability

All data generated and script to analyse them is provided on the dryad repository: http://dx.doi.org/10.5061/dryad.t76fk80.

The following dataset was generated:

| Author(s) | Year | Dataset title | Dataset URL | Database, license, and accessibility information |
|---|---|---|---|---|
| Fanny Pouyet, Simon Aeschbacher, Alexandre Thiéry, Laurent Excoffier | 2018 | Data from: Background selection and biased gene conversion affect more than 95% of the human genome and bias | http://dx.doi.org/10.5061/dryad.t76fk80 | Available at Dryad Digital Repository under a CC0 Public Domain Dedication |

demographic inferences

The following previously published datasets were used:

| Author(s) | Year | Dataset title | Dataset URL | Database, license, and accessibility information |
|---|---|---|---|---|
| Mallick S, Li H, Lipson M, Mathieson I, Gimrek M, et al. | 2016 | Simons Genome Diversity Panel | https://docs.cancergenomicscloud.org/v1.0/docs/simons-genome-diversity-project-sgdp-dataset | Publicly available at Cancer Genomics Cloud (https://www.cancergenomicscloud.org/). |
| Auton A, Brooks RD, Durbin RM, Garrison EP, Kang HM, et al. | 2015 | 1000G Project | ftp://ftp.1000genomes.ebi.ac.uk/vol1/ftp/ | Publicly available at The International Genome Sample Resource (http://internationalgenome.org/home). |

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
