## [Decision Letter]

Thank you for submitting your article "Identifying the neutrally evolving fraction of the human genome" for consideration by *eLife*. Your article has been reviewed by Patricia Wittkopp as the Senior Editor a Reviewing Editor, and three reviewers. The following individual involved in review of your submission has agreed to reveal his identity: Ilan Gronau (Reviewer #3).

The reviewers have discussed the reviews with one another and the Reviewing Editor has drafted this decision to help you prepare a revised submission.

Summary:

In this paper, Pouyet and colleagues develop a new framework to identify to what extent the human genome can be considered "neutral". In particular, the authors define a new statistic, INDA, that should be robust to population demographic, thus allowing the identification of genomic regions where patterns of diversity are not affected by natural selection. They apply this approach to 1000 Genomes and SGDP data. They find a strong positive relationship between local recombination rate and INDA, and that background selection is the primary process that skews genomic diversity, though biased gene conversion also contributes. They determine that WW and SS polymorphisms in regions of the genome where the recombination rate is >1.5cM/MB act effectively neutrally. ~2.5% of SNPs fall into this category, a fairly striking finding that suggests 95% of the genome is affected by natural selection. The authors also show that alternative definitions of neutrality lead to skewed allele frequency spectra which then have downstream affects when attempting to infer demographic history.

Essential revisions:

In general, the three reviewers were complementary of the manuscript and found it to be of value. However, five major issues that required addressing in a revision were identified and agreed upon by all reviewers and the reviewing editor and are summarized below.

1) The current description of INDA in the main text is confusing. The definition should be expanded from its current version in the Results section, and then described in detail in the methods section. Text from Figure 1—figure supplement 1 would be better served being incorporated into the main text Figure 1. The description in Figure 1—figure supplement 1 is also problematic. Stating that the expected value of INDA depends only on T_DIV_ and not on *t*_MRCA_ is a bit of an oversimplification. Under neutrality with no differences in mutation rate, E(INDA) should be the same across populations. However, the probability that a site will be segregating in the set of populations *does* depend on *t*_MRCA_ and therefore differences in *t*_MRCA_ at some loci will influence INDA. Maybe it should be stated that *under neutrality*, E(*t*_MRCA_) is uniform across the genome, however when there is selection, the *t*_MRCA_ can differ at different loci and therefore different genomic regions can have different INDA?

2) Differences in mutation rate across genome will have a big effect on INDA and there is clearly a big difference in mutation rates across the genome. While the authors acknowledge that INDA is affected by mutation rate, they then seem to assume that the entire correlation between INDA and recombination rate is driven by selection. While recombination rate does not seem to be mutagenic based on previous studies, it could still be correlated with something that does affect the mutation rate (e.g. potentially replication timing or transcription). One way to control for mutation rate would be to use divergence from a species such as macaque (or by controlling for trinucleotide context given that CpG sites are already identified as being "peculiar"). As pointed out by Lohmueller et al. (2011), divergence is still affected by BGS, but the effect should be smaller for more distant species. At the very least, the authors need to include more discussion that some of what they see could be driven by mutation rate variation across the genome.

3) After the initial analysis, the neutral sites are identified solely based on recombination rate and mutation pattern (*r*>1.5 cM/Mb, WW,SS). Distance from genes and evolutionary conservation are completely ignored at this stage. The authors should examine whether the resulting neutral set does not have any residual correlations between INDA and other possible covariates, in particular (genetic as well as physical) distance to genes, density of conserved elements and B statistics. It would be surprising if there was no residual correlation with distance to coding exons or other conserved elements (the latter of which has been addressed in several recent studies, e.g. Elyashiv et al., 2016). One reviewer suggests these correlations could be examined on the *r*>1.5 cM/Mb set, though this may lack power due to the restricted size of the "neutral set". Alternatively, another reviewer suggests to use the marginal correlations shown in Figure 1—figure supplement 5 to suggest good cutoffs for the other covariates.

In a similar vein, the authors are assuming that only sites with a local recombination rate > 1.5 will evolve neutrally. However, it seems plausible that sites in low recombination regions could also evolve neutrally if they were separated from selected sites by recombination hotspots, and as such 95% of the genome being neutral may be an overestimate. One reviewer suggests that B values may provide a better estimate? (i.e. sites with B close to 1.0).

4) The authors should comment on the significance of the neutrality threshold occurring at a recombination rate of 1.5cM/Mb. Perhaps background selection is ineffective at recombination rates above 1.5cM/Mb because the human mutation rate is somewhere between 1.0 and 1.5 e-8 mutations per site per generation? Is background selection ineffective in this regime because recombination events occur at a higher rate than mutation events, meaning that mutations are decoupled from each other onto separate haplotypes soon after they occur? If an analytical equation cannot answer this question easily, it could be investigated using *SLiM* by trying out a few different combinations of mutation and recombination rates and seeing whether background selection is effective precisely when the recombination rate is lower than the mutation rate.

5) The lack of a golden standard in terms of neutrality make it very difficult to positively assess that a given set is truly neutrally evolving. This study demonstrates that the author's SNP set is "more neutral" compared to alternative sets that have been determined using fairly simple criteria (Figure 3). While some investigators still use synonymous SNPs, and SNPs far from genes as neutral proxies, there do exist some better ways to filter SNPs (e.g., the NRE of Arbiza et al., 2012, BMC Bioinformatics). The authors should compare their results against these alternative, more sophisticated neutral sets. One reviewer notes that NRE can be used to filter based on B, so the authors can use it to compare their neutral set with the set of regions with very high B.

---

## [Author Response]

Summary:In this paper, Pouyet and colleagues develop a new framework to identify to what extent the human genome can be considered "neutral". In particular, the authors define a new statistic, INDA, that should be robust to population demographic, thus allowing the identification of genomic regions where patterns of diversity are not affected by natural selection. They apply this approach to 1000 Genomes and SGDP data. They find a strong positive relationship between local recombination rate and INDA, and that background selection is the primary process that skews genomic diversity, though biased gene conversion also contributes. They determine that WW and SS polymorphisms in regions of the genome where the recombination rate is >1.5cM/Mb act effectively neutrally. ~2.5% of SNPs fall into this category, a fairly striking finding that suggests 95% of the genome is affected by natural selection. The authors also show that alternative definitions of neutrality lead to skewed allele frequency spectra which then have downstream affects when attempting to infer demographic history.

We appreciate the nice summary of our study. We would just mention that we find that “only” 80-85% of the genome seems affected by selection (and not 95%), whereas about 15% seem affected primarily by gBGC.

Essential revisions:In general, the three reviewers were complementary of the manuscript and found it to be of value. However, five major issues that required addressing in a revision were identified and agreed upon by all reviewers and the reviewing editor and are summarized below.

*1) The current description of INDA in the main text is confusing. The definition should be expanded from its current version in the Results section, and then described in detail in the methods section. Text from Figure 1—figure supplement 1 would be better served being incorporated into the main text Figure 1. The description in Figure 1—figure supplement 1 is also problematic. Stating that the expected value of INDA depends only on T_DIV_ and not on t_MRCA_ is a bit of an oversimplification. Under neutrality with no differences in mutation rate, E(INDA) should be the same across populations. However, the probability that a site will be segregating in the set of populations* does *depend on t_MRCA_ and therefore differences in t_MRCA_ at some loci will influence INDA. Maybe it should be stated that* under neutrality*, E(t_MRCA_) is uniform across the genome, however when there is selection, the t_MRCA_ can differ at different loci and therefore different genomic regions can have different INDA?*

We have followed the suggestions of the reviewers. We have modified the Results section to better introduce this statistic and justify its use and we have considerably extended its description in the Material and methods section to clarify its definition, hoping it removes any ambiguity. Note that in order to follow the suggestion of the third reviewer, we have dropped the use of INDA in favor of the average derived allele frequency, noted as DAFi¯, which is just INDA divided by twice the number of polymorphic sites in the sample, as it is easier to compare this statistic across data sets. In the new version, INDA, now simply called *n_i_*, is only described in the material and methods to show its quasi-independence of differential demography among populations. DAFi¯ is used in the reminder of the paper, and all figures have been modified to reflect this change.

2) Differences in mutation rate across genome will have a big effect on INDA and there is clearly a big difference in mutation rates across the genome. While the authors acknowledge that INDA is affected by mutation rate, they then seem to assume that the entire correlation between INDA and recombination rate is driven by selection. While recombination rate does not seem to be mutagenic based on previous studies, it could still be correlated with something that does affect the mutation rate (e.g. potentially replication timing or transcription). One way to control for mutation rate would be to use divergence from a species such as macaque (or by controlling for trinucleotide context given that CpG sites are already identified as being "peculiar"). As pointed out by Lohmueller et al. (2011), divergence is still affected by BGS, but the effect should be smaller for more distant species. At the very least the authors need to include more discussion that some of what they see could be driven by mutation rate variation across the genome.

We agree that there is a wide difference in mutation and recombination rates along the genome, and that these two processes might be directly (mutagenic effect of recombination) or indirectly (e.g. through replication timing) correlated. To address these concerns in more details, we have now added a full paragraph on the possible mutagenic effect of recombination at the beginning of the Discussion section as well as on the possible indirect association between mutation rate and recombination. We have also added a new analysis of the relationship between the intensity of BGS quantified by McVicker et al. (2009) in terms of the B-statistic and the recombination rate under a simple model. We show that the fit to the data is significantly better if we allow for some correlation between mutation and recombination rates. Since the sole mutation–recombination correlation cannot explain the exact relationship observed between DAFi¯ and recombination (i.e. the plateau for DAFi¯ above ~1.5 cM/Mb for SS+WW sites) and the difference in the shape of the SFS for low and high recombination regions, we now conclude that our observations are best explained by the action of BGS and gBGC and by the presence of some correlation between mutation and recombination along the genome. We have modified the Discussion and Material and Methods section to summarise our conclusions and describe the additional analyses (see also the new Figure 1—figure supplement 8 and Supplementary file 4).

3) After the initial analysis, the neutral sites are identified solely based on recombination rate and mutation pattern (r>1.5 cM/Mb, WW,SS). Distance from genes and evolutionary conservation are completely ignored at this stage. The authors should examine whether the resulting neutral set does not have any residual correlations between INDA and other possible covariates, in particular (genetic as well as physical) distance to genes, density of conserved elements and B statistics. It would be surprising if there was no residual correlation with distance to coding exons or other conserved elements (the latter of which has been addressed in several recent studies, e.g. Elyashiv et al., 2016). One reviewer suggests these correlations could be examined on the r>1.5 cM/Mb set, though this may lack power due to the restricted size of the "neutral set". Alternatively, another reviewer suggests to use the marginal correlations shown in Figure 1—figure supplement 5 to suggest good cutoffs for the other covariates.In a similar vein, the authors are assuming that only sites with a local recombination rate > 1.5 will evolve neutrally. However, it seems plausible that sites in low recombination regions could also evolve neutrally if they were separated from selected sites by recombination hotspots, and as such 95% of the genome being neutral may be an overestimate. One reviewer suggests that B values may provide a better estimate? (i.e. sites with B close to 1.0).

We found it indeed interesting and important to better characterize our neutral set. We have done so by examining, for our neutral set of SNPs, possible relationships between DAFi¯ (now replacing INDA) and other covariates, such as the B-statistics as well as the distance (in terms of map units) to exons, to conserved elements and to recombination hotspots. In addition, we have compared the SFS observed in our neutral set (SS+WW sites with *r* > 1.5 cM/Mb) to the SFS computed for subsets of SNPs that showed extreme values for the covariates mentioned above (first and fourth quartile). In our neutral set, we find virtually not relationship between DAFi¯ and recombination rate or little for distance to exons (DAFi¯ varies from 0.145 to 0.15, close to the average of 0.146), but we find a negative relation with distance to recombination hotspots and a positive relationship with distances to conserved elements and with B-statistics. We find that the distance to phastCons conserved elements has a major influence on DAFi¯. However, removing the 25% closest sites to conserved elements sites has little effect on the SFS (the fourth quartile is close to the whole set suggesting that a majority of SNPs behave as the fourth quartile and not as the first one), suggesting that our whole neutral set can be used for SFS-based demographic inference, even though it may appear safer to remove these sites from some analyses. These results are reported in Figure 1—figure supplement 9 and Figure 2—figure supplement 3 and discussed in a new paragraph of the Discussion section. We also added a few sentences to acknowledge the fact that regions with low recombination rates can still harbor sites unaffected to BGS, but that the average DAFi¯ statistic has not enough resolution to recognize these regions.

4) The authors should comment on the significance of the neutrality threshold occurring at a recombination rate of 1.5cM/Mb. Perhaps background selection is ineffective at recombination rates above 1.5cM/Mb because the human mutation rate is somewhere between 1.0 and 1.5 e-8 mutations per site per generation? Is background selection ineffective in this regime because recombination events occur at a higher rate than mutation events, meaning that mutations are decoupled from each other onto separate haplotypes soon after they occur? If an analytical equation cannot answer this question easily, it could be investigated using SLiM by trying out a few different combinations of mutation and recombination rates and seeing whether background selection is effective precisely when the recombination rate is lower than the mutation rate.

We agree that a better theoretical justification of the threshold recombination rate used to define neutral regions would be of value. Actually, previous theory shows that the ratio of the deleterious mutation rate to the recombination rate is a major determinant of the effect of background selection (Hudson and Kaplan, 1995; Nordborg et al., 1996). We have therefore addressed this concern by using the observed linear relationship between DAFi¯ and B-statistics for which a dependence on BGS and recombination is theoretically available, and we explicitly used it to address point 2 above. We have thus used a simple model of BGS (Hudson and Kaplan, 1995) describing the relationship between B and recombination rate to evidence a potential correlation between recombination and mutation. In this analysis, we find that the B-statistic also reaches a plateau approximately above 1.5 cM/Mb by assuming a log-log linear relation between the deleterious mutation rate and recombination. In this simple BGS model, B*≈* exp(u*_d_*/r), where u*_d_* is the deleterious mutation rate. Thus, B tends symptotically to 1 with increasing r values, but more slowly when u*_d_* increases with r than when it is constant. In this analysis, in order to correctly explain the data, we find the deleterious mutation rate to be about 10 times smaller (∼1.4e-9 per site per generation) than the neutral mutation rate when r= 1 cM/Mb (see Figure 1—figure supplement 8). Of course, this model is very simple, but we think it predicts relatively well the main features of the average B statistics. Given that average DAFi¯ and B-statistics are highly correlated, the threshold recombination rate of 1.5 cM/Mb should also apply to DAFi¯. Therefore, the intuition of the reviewers is correct in the sense that there is a theoretical justification for this plateau, but theory suggests that it depends on the interaction between the deleterious mutation rate and recombination, and not between the neutral mutation rate and recombination.

Because the analytical theory provides a sharp qualitative prediction that justifies our threshold of 1.5 cM/Mb, we decided not to run additional simulations. We do, however, stress that not too much emphasis should be given to the exact value of 1.5 cM/Mb, but rather to the claim that the true threshold is expected to be roughly around this recombination rate.

5) The lack of a golden standard in terms of neutrality make it very difficult to positively assess that a given set is truly neutrally evolving. This study demonstrates that the author's SNP set is "more neutral" compared to alternative sets that have been determined using fairly simple criteria (Figure 3). While some investigators still use synonymous SNPs, and SNPs far from genes as neutral proxies, there do exist some better ways to filter SNPs (e.g., the NRE of Arbiza et al., 2012, BMC Bioinformatics). The authors should compare their results against these alternative, more sophisticated neutral sets. One reviewer notes that NRE can be used to filter based on B, so the authors can use it to compare their neutral set with the set of regions with very high B.

This is a good comment and to address it, we have also compared our neutral set to one that was initially defined by Gronau et al. (2011), and which consists of a series of 37,574 1-kb regions cleared of misalignment with chimpanzees, without CpG sites, exonic regions, or any conserved non-coding elements, at least 50 kb away from each other, without recombination hot-spots within regions and separated by hot-spots to minimize their linkage. These regions potentially minimally affected by BGS have been used as a neutral benchmark in several papers referenced in the discussion. We have also compared our neutral set to regions associated with B-statistics larger than 0.9, as suggested by one reviewer. These two alternative data sets show clearly different SFS shapes than our neutral set, mainly due to BGS for the 37,574 1-kb regions and to gBGC for the B > 0.9 SNPs. These alternative data sets are discussed in the Discussion section and their SFS are presented in new Figure 3—figure supplement 4A. The B > 0.9 data set including only WW and SS SNPs could therefore be considered as a good alternative data set on which to perform demographic inference.